# Effect of Active Pedagogical Models on Basic Psychological Needs and Intention to Do Physical Exercise

**DOI:** 10.3390/bs15111574

**Published:** 2025-11-17

**Authors:** Olga Calle, Antonio Antúnez, Sergio José Ibáñez, Sebastián Feu

**Affiliations:** 1Training Optimization and Sport Performance Research Group (GOERD), Sport Science Faculty, University of Extremadura, 10005 Caceres, Spain; antunez@unex.es (A.A.); sibanez@unex.es (S.J.I.); sfeu@unex.es (S.F.); 2Faculty of Sport Sciences, University of Extremadura, 10003 Caceres, Spain; 3Faculty of Education and Psychology, University of Extremadura, 06006 Badajoz, Spain

**Keywords:** basic psychological needs, intention to do physical exercise, pedagogical models, physical education, alternative invasion sport, gender

## Abstract

(1) This study examined the influence of pedagogical models and gender on basic psychological needs and intention to engage in physical exercise during the teaching of an alternative invasive sport. (2) Participants were 136 students from the fifth and sixth grades of Primary Education and the first grade of Secondary Education (*M* = 11.36; *SD* = 1.04). Two programs were implemented per grade: one following the Game-Centered Model, and another based on a hybrid model combining the Game-Centered Model and the Sports Education Model. Autonomy, competence, and relatedness were assessed using the Basic Psychological Needs in Physical Exercise Scale, while exercise intention was evaluated with the Intentionality to Be Physically Active Scale. Psychometric properties were verified through confirmatory factor analysis, Cronbach’s alpha, mean variance extracted, and composite reliability. Descriptive and inferential analyses were conducted using the linear mixed model and Bonferroni’s post hoc test. (3) Both pedagogical models improved all variables. The hybrid model yielded significantly higher autonomy. Gender differences were observed in relatedness, with boys reporting greater values. (4) The pedagogical models used favor the satisfaction of basic psychological needs and exercise intention. Teachers should integrate psychological needs into programs for improvements in intention to be physically active, positively impacting self-determined participation.

## 1. Introduction

Physical inactivity produces negative consequences for physical and mental health ([48]), reaching higher levels in children and adolescents every day according to the [89] ([89]). This circumstance derives from contextual factors of contemporary society in which technologies and sedentarism predominate ([19]). Invasion sports are one of the physical education (PE) contents most used by teachers in Spain, being an ideal scenario for the promotion of physical activity ([66]). Invasion sports are team-based sports in which two teams compete simultaneously within a defined and shared playing area, following established rules and using a game object (ball, disk, among others) to penetrate the opponent’s territory and score points, while concurrently defending their own area. These sports are characterized by the necessity for continuous cooperation among teammates, opposition against the rival team, and tactical decision-making within dynamic and unpredictable contexts, which demand adaptation, effective communication, and comprehensive game understanding ([66]; [70]; [88]).

Currently, alternative sports are emerging as an approach with great educational potential in the context of physical education ([57]). These disciplines contribute to the integral development of students ([5]), favoring access, participation, and training due to their novel, unknown, and mixed characteristics ([4]; [56]). “Rosquilla” is an alternative invasion sport created by Professor Manuel Rodríguez-Barriga, played on a rectangular field that includes two circular areas designated for scoring points. The sport uses a PVC hoop with a diameter of 28 cm. Two teams, each consisting of 4 or 5 players, compete with the objective of scoring more points than the opposing team. To score, the coordinated action of two players from the same team is required; an attacking player must throw the hoop into the circular attack area, which is occupied by another teammate, who must then insert the hoop onto their arm and subsequently place it on the ground within the area. Matches consist of three 10 min periods with a 5 min break. Play begins with a jump between two players from opposing teams at the center of the field. The specific rules include the following. (i) The player holding the hoop may take no more than two steps and cannot hold the hoop for more than 5 s; (ii) a defending player may not come closer than 1 m to the attacking player with the ring; (iii) an attacking player may remain in the circular attack area for no more than 5 s, and if this is exceeded, play resumes with a sideline throw by the opposing team; (iv) defending players may not enter the circular area they are guarding, with violations penalized by two direct throws; (v) if the ring falls to the ground, play resumes with a sideline throw by the team that did not lose possession; (vi) points cannot be scored directly from a sideline throw. The pedagogical objectives of “Rosquilla” include promoting equal participation and coeducation through mixed-gender teams, stimulating cooperation and teamwork, reducing the technical skill demands on players, and providing adaptability in terms of space and materials ([73]). Figure 1 shows the fundamental characteristics of the alternative invasion sport, “Rosquilla”.

Sport-based learning, particularly through invasion games and alternative invasion sports, provides an optimal context for promoting engagement, cooperative interaction, and decision-making, all of which are closely associated with motivational processes. These activities offer students opportunities to take initiative, collaborate effectively, and face meaningful challenges that foster personal development, thereby laying the groundwork for the satisfaction of basic psychological needs and the development of self-determined motivation. Self-determination theory focuses on motivation and personality, emphasizing the ability of subjects to regulate their own behaviors to be responsible for their actions ([74]). It includes the theory of basic psychological needs (BPN) ([17]). BPN contain three dimensions of people’s development: (1) autonomy (refers to the self-regulation of behaviors by the subject, involving the voluntary choice of activities and goals through decision making); (2) competence (refers to the individual’s perception of the level of skill and effectiveness in performing tasks); (3) relatedness (feeling of connection, belonging and bonding with the people around him/her and with the community) ([17]).

The satisfaction of the three dimensions of BPN is related to motivation ([74]), so that if individuals feel that they make decisions, are efficient, and form optimal relationships with their peers, greater intrinsic motivation is obtained ([84]), an aspect that enhances their participation and self-determined regulation, which leads pupils to practice a sport modality on their own initiative ([74]). On the contrary, if people do not satisfy these needs, the activity is not valued intrinsically ([84]), which can generate extrinsic motivation (participating to obtain a reward or avoid punishment) or demotivation (which sometimes leads to abandonment of the sport) ([83]). The satisfaction of BPN in physical education is key to achieving meaningful learning and promotion of, as well as adherence to, physical exercise ([21]), while frustration of BPN leads to a decrease in sport practice as the educational level increases ([49]). This theory allows an understanding of the motivational process of students to be formed ([84]). Within this framework, understanding how sports learning processes and methodological approaches impact the satisfaction of students’ BPN becomes crucial, given their significant influence on students’ participation and motivation.

The Sport Education Model (SEM) was developed by Siedentop ([78]). Currently, it is very well accepted in Spain ([40]), as it provides a competence-based education, whose objective is that students have a real experience of sport ([78]). Basically, it relies on six structural elements of sport that are adapted to the educational context, such as team affiliation, seasons, formal competition, record keeping, festivity, and a culminating event ([78]). These phases are accompanied by the performance of roles, functions, and responsibilities, favoring the experience and knowledge of sport ([78]).

The SEM promotes the satisfaction of BPN ([15]), due to its structural elements ([69]), which leads to higher levels of intrinsic motivation ([68]; [80]) and self-determined motivation ([68]), positively correlated with student participation or active engagement ([61], [62]). In this line, improvements have been found in autonomy and self-determined behaviors ([51]; [86]), in the perception of competence and success ([51]; [80]), and in relatedness and socialization ([68]).

The Game-Centered Model (GCM) ([43]) is aimed at the tactical-technical development of the sport. It is oriented to the management of fundamentals and practical applications that lead to the mastery of knowledge, content, and sport skills to achieve meaningful learning in contextualized game situations, developing great autonomy ([38]). Some studies in Spain attest that this model satisfies students’ BPN, leading to the achievement of self-determined motivation ([32]; [31]; [79]), and that this model generates greater autonomy, competence, and satisfaction compared to traditional methodology ([35]). In the international context, it has been shown that pedagogical models such as SEM and GCM promote both participation and satisfaction of BPN, leading to the enhancement of active learning, which provides benefits in social, physical, and cognitive domains ([18]).

Hybrid models emerge to adapt to educational needs and improve the functionality of the models ([37]), being accepted by the specialized literature ([25]). The hybrid model combining the GCM and the SEM (HM) provides improvements in game understanding, skill development, and student motivation ([20]; [35]; [37]). Under these premises, we question whether the implementation of active methodologies such as GCM and HM positively affects the satisfaction of BPN, with the aim of enabling educators to regulate BPN through the application of these pedagogical models. The HM of the GCM and SEM introduces phases of shared leadership (between teacher and students) and student autonomy, in which students assume roles and responsibilities. In this context, we question whether these characteristics inherent to the pedagogical model provide greater satisfaction of the autonomy dimension of BPN compared to the GCM, where leadership rests exclusively with the teacher.

Adherence to physical activity corresponds to being physically active ([47]). Adherence to physical activity throughout life requires starting from adequate practice, inculcating the need for practice and the benefits from an early age ([81]). The theory of planned behavior states that attitude influences intention ([1]). [67] ([67]) found that a positive attitude towards sports practice is a positive predictor of behavior. PE classes constitute a space of initial contact with physical activity ([41]), in which students experience the benefits of practice ([12]). Therefore, the positive or negative attitudes they experience both inside and outside the school context are determinants for developing lasting physical exercise habits over time. Therefore, it is necessary to motivate people to partake in physical activity to foster the intention to be physically active, which favors adherence to the practice ([2]). With respect to the SEM, [86] ([86], [85]) prove the existing relation between motivation with the intention to exercise, which produces transfer to behaviors. It has been shown that personal influences and pedagogical models affect students’ motivation and learning outcomes in physical education, thereby promoting the intention to engage in physical activity ([10]). From a pragmatic perspective, it is necessary to examine how the implementation of these active pedagogical models affects the intention to engage in physical exercise. Specifically, we ask whether there is a positive relationship between basic psychological needs (BPN) and the intention to exercise in our specific context, so that educators can design interventions that satisfy BPN and, consequently, enhance students’ intention to participate in physical activity.

Invasion sports bring greater satisfaction of BPN and adherence to exercise in boys ([23]), and there is a predisposition to unequal and stereotyped participation ([39]). In this field, physical inequalities are evident, an aspect that generates less participation from girls and less skilled students ([53]; [76]). However, there is no evidence on how the practice of invasive alternative sports influences these variables through the application of specific pedagogical models. As established by [53] ([53]), an adequate coeducational didactic approach is required for equity in alternative sports participation to occur. Therefore, the role of the teacher is key in achieving coeducational treatment and promoting the equal participation of students ([28]). As emphasized in previous research, it is crucial to examine whether the satisfaction of basic psychological needs (BPN) and the intention to participate in physical activity vary according to gender when learning an invasion sport, or whether, instead, offering a mixed-gender sporting context through this alternative sport facilitates comparable BPN satisfaction across genders.

It is essential to know how methodologies affect the satisfaction of BPN and the intention to be physically active, which may favor intrinsic motivation, learning, and the quality of the educational system. However, there is limited research on how pedagogical models affect psychological needs in alternative sports such as “Rosquilla”. This study is the first to explore these effects within this unique context. Therefore, the aim of this study was to analyze the influence of pedagogical models and gender on psychological variables (BPN and intention to do physical exercise) during the learning of an alternative sport, “Rosquilla”, in the school context. The hypotheses were that (1) students will show greater satisfaction with BPN and intention to do physical exercise after the intervention programs, (2) the HM will obtain higher levels of autonomy compared to the GCM, (3) BPN will correlate positively with intention to do physical exercise, and (4) boys will show greater satisfaction across all dimensions of basic psychological needs (autonomy, competence, and relatedness) as well as a higher intention to engage in physical activity than girls.

## 2. Materials and Methods

### 2.1. Study Design

The study is framed within a mixed factorial quasi-experimental research design with two groups of repeated measures (pre-test/post-test) ([63]). Two intervention programs were developed and validated ([7]), in which students experienced for the first time both the pedagogical models, GCM and the HM of GCM and SEM, and the alternative sport, “Rosquilla”. The interventions had a duration of two months in each case.

### 2.2. Participants

In the present study, non-probability sampling was used for the selection of students under the category of participating subjects due to convenience ([63]), motivated by aspects of accessibility of the researchers. The sample included 136 students (74 girls and 62 boys), divided into six groups of the fifth and sixth year of Primary Education and first year of Secondary Education (*M* = 11.36; *SD* = 1.04), from a public high school and a public school in southwestern Spain. Six natural groups from the Spanish educational system were selected. Within this system, the educational administration organizes groups according to pedagogical and equity criteria, ensuring mixed and heterogeneous configurations. Therefore, the criteria established by the educational administration regarding group composition were respected to guarantee the ecological validity of the study. The pedagogical models were randomly assigned to the different groups. Students had no prior experience with the sport “Rosquilla” or with the pedagogical models implemented, which allowed all students to start at the same level for practicing “Rosquilla” and facilitated the control of pre-existing differences between groups and students. However, the results may still be influenced by the natural structure of the research groups.

The inclusion criteria were as follows: (i) informed consent of the legal guardians; (ii) the students had to have participated in at least 80% of the sessions; (iii) they had to complete all the items in the questionnaires. Figure 2 presents the characteristics of the sample for this study.

### 2.3. Variables and Instruments

The dependent variables were as follows: (i) BPN (autonomy, competence, and relatedness); (ii) intention to do physical exercise. The independent variable was the type of intervention program (GCM; the HM of the GCM and the SEM). Gender (male and female) was included as a grouping variable (non-manipulated).

The intervention programs were analogous in terms of objectives, type of tasks, contents, and phases of the game. They were differentiated by the characteristics of each pedagogical model (Figure 3, extracted from [6]). A panel of 9 expert judges validated the interventions, with experience in pedagogical models and invasive sports. The programs achieved excellent content validity (Aiken’s V ≥ 0.73) and internal consistency (Cronbach’s α = 0.99) ([7]). In the HM, students have more autonomy. The tasks of the GCM are led by the teacher, while HM tasks are led by the teacher during the affiliation phase, and in the first sessions of the season phase, students lead the sessions during the final sessions of the season phase, as well as the competition, registration, and final event phases, in which the teacher guides them.

Data were obtained before and after applying the two programs based on each of the pedagogical models by means of a series of questionnaires validated and accepted by the scientific community. The students completed two instruments. The first was the (i) Basic Psychological Needs Scale (BPNES), adapted to Spanish and validated by [60] ([60]). This instrument consists of 12 items grouped into 3 subscales (autonomy, competence and relationship with others) containing 4 items each, using a Likert-type scale ranging from 1 (strongly disagree) to 5 (strongly agree) ([60]). The second was the (ii) measure of intentionality to be physically active (MIPA), containing five items using a Likert-type scale ranging from 1 (strongly disagree) to 5 (strongly agree) ([2]).

### 2.4. Procedure

The objectives of the research were explained to the school management, the teaching staff, and the legal guardians of the students beforehand. Approval was gained from the University’s Bioethics and Biosafety Committee (159/2022). In addition, the authorization and informed consent required for the study were acquired. All procedures outlined in the Declaration of Helsinki ([90]), in relation to the Social Sciences, were followed, guaranteeing the anonymity of the participants.

The research was carried out in five phases. In the first phase, all permits were obtained. In the second phase, an initial assessment (pre-test) was conducted. Then, two programs (12 sessions) were implemented for two months. The two interventions were conducted by a researcher and a teacher, both experts in pedagogical models and the alternative sport, with the intention of reducing variability in the interventions, supported by a group of specialists who advised and supervised the interventions. A detailed record of each session and task was maintained, allowing for the correction of errors and the appropriate adjustment of the two programs based on the pedagogical models, always with the guidance and support of the expert group. The programs were implemented as designed and planned. Subsequently, the final evaluation (post-test) was applied. Finally, the databases for analysis were created. Figure 4 includes the procedure developed in this study.

### 2.5. Statistical Analysis

The factor model of the two scales used in the study was evaluated through confirmatory factor analysis, and the internal consistency was assessed using Cronbach’s alpha ([26]). Descriptive data were calculated using the mean and standard deviation to obtain BPN scores and the intention to exercise in both the pre-test and post-test for each of the students. Assumption tests were conducted to identify the characteristics of the study data ([26]). Data normality was assessed separately for each measurement point (pre-test and post-test) using the Kolmogorov–Smirnov test. Results indicated significant deviations from normality for all variables: (i) autonomy pre-test (*D* = 0.200; *p* < 0.001) and post-test (*D* = 0.185; *p* < 0.001); (ii) competence pre-test (*D* = 0.172; *p* < 0.001) and post-test (*D* = 0.135; *p* < 0.001); (iii) relatedness pre-test (*D* = 0.124; *p* < 0.001) and post-test (*D* = 0.140; *p* < 0.001); (iv) intention pre-test (*D* = 0.130; *p* < 0.001) and post-test (*D* = 0.140; *p* < 0.001). Since none of the distributions met the normality assumption, non-parametric tests were applied. However, when the variable distributions violate normality, the linear mixed model can still be used ([87]). Therefore, a linear mixed model (LMM) and a Bonferroni post hoc test ([46]) were performed to identify statistical differences between the variables studied (BPN and intention to exercise) with respect to methodology and gender. The LMM reflects the variability of participants’ individual responses. The Bonferroni post hoc multiple comparison test tested for differences between groups according to methodology, gender, and the interaction of these variables ([26]). Spearman’s rank correlation coefficient was also used to find relationships between BPN (autonomy, competence, and relatedness) and intention to exercise. Differences were considered significant when *p* ≤ 0.05.

In addition, the effect size, partial eta squared (*η*^2^), and the observed power (*ϕ*) were estimated. On the one hand, the effect size *η*^2^ was classified as small (0.010–0.059), medium (0.060–0.139), and large (>0.140). On the other hand, the observed power values (>0.80) were interpreted as optimal ([9]).

Finally, the analyses were completed with Jamovi software version 2.3.24 (The Jamovi Project, Sydney, Australia, 2022) and Statistical Package for the Social Sciences (SPSS) software version 25 (IBM Corp. 2012. IBM SPSS Statistics for Windows, IBM Corp., Armonk, NY, USA).

## 3. Results

The items of the BPNES and MIPA questionnaires were analyzed by means of confirmatory factor analysis. In the BPNES questionnaire, after a first analysis, it was necessary to use index modification, relating the following residual covariance items 6 and 9, items 5 and 8, items 4 and 7, and items 3 and 9, ensuring that these were acceptable: CMIN/DF = 2.23, CFI = 0.98; TLI = 0.97; SRMR = 0.03, RMSEA = 0.07 ([45]). In the MIPA questionnaire, after a first analysis, it was necessary to use index modification, as the residual covariances of items 1 and 2 were related, to ensure that these were acceptable: CMIN/DF = 1.92, CFI = 1.00; TLI = 0.98; SRMR = 0.01, RMSEA = 0.06. The BPNES (*AIC* = 7283.18; *BIC* = 7438.23) presented reliability in its three factors—satisfaction with BPN, autonomy (*α* = 0.93), competence (*α* = 0.88), and relatedness with others (*α* = 0.88)—while the MIPA (*AIC* = 3031.69; *BIC* = 3089.38) showed optimal reliability (*α* = 0.90) ([26]).

The descriptive results of the variables, BPN, and intention to exercise, for each intervention program and gender, are presented in Table 1.

The inferential results of the psychological variables analyzed for the factors of moment and pedagogical model are presented in Table 2. There were statistically significant differences in the within-subject factor for autonomy (*F* = 846.3; *p* < 0.001; *n*^2^ = 0.86), for competence (*F* = 477.71; *p* < 0.001; *n*^2^ = 0.78), for relatedness (*F* = 342.24; *p* < 0.001; *n*^2^ = 0.72), and for intention (*F* = 1029.55; *p* < 0.001; *n*^2^ = 0.88). There were statistically significant differences in the inter-subject factor for autonomy (*F* = 16.2; *p* < 0.001; *n*^2^ = 0.11). Likewise, there were significant differences in the moment x model interaction for autonomy (*F* = 121.6; *p* < 0.001; *n*^2^ = 0.48), for competence (F = 30.33; *p* < 0.001; *n*^2^ = 0.18), for relatedness (*F* = 74.58; *p* < 0.001; *n*^2^ = 0.36), as well as for intention (*F* = 6.27; *p* = 0.01; *n*^2^ = 0.004). The high effect sizes (*η*^2^ = 0.72–0.88) indicate that the intervention accounts for 72% to 88% of the variance in autonomy, competence, relatedness, and intention, reflecting a highly significant impact. In the between-subjects factor, a moderate effect was observed only for autonomy (*η*^2^ = 0.11), suggesting that the HM significantly enhances autonomy compared to the GCM. Moreover, significant moment × model interactions were found, with moderate-to-large effects for autonomy (*η*^2^ = 0.48), competence (*η*^2^ = 0.18), and relatedness (*η*^2^ = 0.36), indicating that pre–post changes differ depending on the model. Practically, these results confirm a robust impact of the interventions and differential effects according to the model, underscoring the importance of designing interventions that target key psychological variables in practice and training contexts.

Table 3 presents the results of controlling for the random effect of each group for the time and model factors. The results showed improvements in the conditional R^2^ compared to the marginal R^2^ for all four variables, indicating that the random effects (group) explain a substantial portion of the variability in participants’ responses. Furthermore, all variables were significant, suggesting that controlling for between-subject variability is necessary. A high intraclass correlation coefficient (*ICC* > 0.5) was obtained, reinforcing that there are considerable differences between groups, which implies that individual differences among students substantially affect the observed outcomes. These findings indicate that both the time × model factors and the group structure have a significant influence on the satisfaction of BPN and participant adherence (*p* < 0.001 in all cases). These results have direct implications for the educational context, as they arise from real teaching situations. Differences between groups and the effects of factors such as time and model reflect how classroom dynamics influence BPN satisfaction and the intention to engage in physical activity.

The inferential results of the psychological variables analyzed for the time and gender factors are recorded in Table 4. There were statistically significant differences in the within-subject factor for autonomy (*F* = 425. 53; *p* < 0.001; *n*^2^ = 0.76), for competence (*F* = 378.63; *p* < 0.001; *n*^2^ = 0.74), for relatedness (*F* = 209.91; *p* < 0.001; *n*^2^ = 0.61), and for intention (*F* = 970.13; *p* < 0.001; *n*^2^ = 0.88). There were statistically significant differences in the inter-subject factor for relatedness (*F* = 8.91; *p* = 0.003; *n*^2^ = 0.01). The analyses revealed large within-subject effects across all evaluated variables, with high effect sizes (*η*^2^ = 0.61–0.88), indicating substantial improvements following the intervention. In contrast, between-subject effects associated with gender were significant only for relatedness, with a small effect size (*η*^2^ ≤ 0.01), suggesting that gender accounts for only a small proportion of the variance, and no significant moment × gender interactions were observed. Practically, these results suggest that the intervention had a robust and generalized impact on autonomy, competence, and intention, regardless of participants’ gender, with minor differences in relatedness, which were greater among males. A lower sense of relatedness can lead to reduced self-esteem and motivation, affecting participation, athletic performance, and engagement, thereby reinforcing gender inequalities. Therefore, it is necessary to implement adjustments in interventions to strengthen girls’ sense of belonging and interpersonal connections.

The results of controlling for the random effect of each group for the time and gender factors are presented in Table 5. Improvements in the conditional R^2^ compared to the marginal R^2^ were observed for all four variables, indicating that, when controlling for the random effect of group, the time and gender factors accounted for a significant proportion of the variance in basic psychological needs (BPN) and the intention to engage in physical activity. Once again, all variables were significant, confirming the necessity of controlling for variability among participants. A high intraclass correlation coefficient (*ICC* ≥ 0.5) was obtained, emphasizing the existence of differences between groups and suggesting that individual differences among participants influence the outcomes. Overall, these findings indicate that both gender and the time point significantly affect BPN and intention, although group context continues to play a relevant role in explaining variance. Given that this study was conducted in a real teaching setting, it should be noted that both between-group differences and the effects of factors such as time and gender reflect how classroom dynamics and individual student characteristics influence the satisfaction of BPN and the intention to engage in physical activity.

All correlations between the psychological variables were significant, as shown in Table 6. The strongest relations were obtained between the variables of competence and autonomy (*p* < 0.001) and competence and relatedness (*p* < 0.001). Significant relations were observed between intention and autonomy (*rs* = 0.56; *p* < 0.001), intention and competence (*rs* = 0.61; *p* < 0.001), and intention and relatedness (*rs* = 0.45; *p* < 0.001).

Figure 5 illustrates the relationship between the variables and the time points according to the pedagogical model, based on the results obtained.

Figure 6 shows the relationship between the variables and the time points according to gender, based on the results obtained.

## 4. Discussion

The aim of the study was to analyze whether pedagogical models and gender influence the variables of satisfaction with BPN and intention to be physically active during the teaching of an alternative sport. Likewise, a test was conducted to identify relations between the psychological variables. The results show improvements in all variables with respect to the models and to gender for each subject. Therefore, we can affirm that hypothesis 1 is confirmed, because students show greater satisfaction in BPN (autonomy, competence, and relatedness) and intention to exercise after the application of the interventions. The results show that there is a correlation between BPN and the intention to be physically active, so hypothesis 3 is accepted. Finally, it is observed that there are significant differences in the relatedness variable according to the gender of the students (boys > girls), so hypothesis 4 is partially accepted, since no significant differences are obtained according to gender for the rest of the variables.

Participants achieve higher BPN satisfaction and intention to exercise with the two models. These results are consistent with some studies showing that the application of an MH-based program produces improvements in BPN satisfaction ([35], [34]; [54]; [82]); the GCM also favors BPN ([32]; [31]; [79]). However, the literature indicates that these models do not consistently produce improvements across all BPN dimensions. According to [55] ([55]), the implementation of an HM, which incorporates the SEM, yields significant improvements only in competence and relatedness, with no significant changes in autonomy. In contrast, [33] ([33]) reported that the HM contributes to improvements in both autonomy and competence. Other studies have observed positive effects only on competence satisfaction when implementing the SEM ([15]; [71]). Furthermore, the review conducted by [65] ([65]) indicates that the effects of the GCM on psychosocial variables are inconsistent, highlighting the need for further studies to evaluate these effects. Satisfaction of BPN leads to self-determined motivation, which promotes greater physical exercise ([77]), which correlates with active participation and engagement ([61], [62]). According to the literature, the satisfaction of BPN drives higher levels of intrinsic motivation ([68]; [80]). This increased motivation fosters self-determined participation ([74]; [68]). In this sense, the work of [86] ([86]) attests to improvements in the motivation of students thanks to the application of the SEM, an aspect that has an impact on greater participation ([85]; [68]). The implementation of appropriate methodologies positively influences students’ motivation, increasing the probability of engaging in physical activity ([10]).

Likewise, the two models produce improvements in the intention to exercise variable. In this line, the continued use of the SEM causes positive effects on the intentionality of doing physical exercise ([64]), with this intentionality being a predictor of actual physical activity ([86], [85]). On the other hand, the study by [59] ([59]) determines that the use of the GCM leads to improvements in sports adherence. These findings contrast with the data reported by [58] ([58]), in which the implementation of the GCM did not lead to improvements in the intention to engage in physical activity. In addition, the application of the hybrid model between the GCM and SEM increased the intention to be physically active in both boys and girls ([50]). We must consider that the heterogeneity of the student body within the PE groups results in the satisfaction of the BPN and the intention not being observed in the same way for all students ([29]). The role of the teacher is fundamental to achieving satisfaction of BPN ([27]), and the teacher’s orientation conditions the environment, classroom climate, and motivation. Also, the pedagogical model is a key element; the scientific community shows that technical models based on reproduction versus active models provide lower levels of motivation ([14]). Therefore, active methodologies should be applied to favor BPN and intention to do physical exercise, which are linked to motivation.

Better scores are obtained from the HM in autonomy compared to the GCM. The characteristics of the SEM favor autonomy ([78]; [30]), due to the implementation of phases in which students work autonomously and there is the inclusion of roles, functions, and responsibilities ([78]). These characteristics largely justify the better scores obtained by the HM in autonomy compared to the GCM, since the GCM is supported by teacher-directed tasks, while the HM proposes teacher-directed tasks, tasks with shared direction between teacher and students, as well as autonomous tasks. Given that autonomy is a transversal element of the educational standards in Primary Education, we should prioritize methodologies that enhance autonomy, such as the SEM or the HM, over other pedagogical models.

When a new pedagogical model is applied, both students and teachers require a period of pedagogical transition to adapt to the demands of the new model, influencing student motivation ([44]); this circumstance is magnified by the limited time provided by an isolated program to generate this adaptation. The findings of this study reflect this need for pedagogical transition, which justifies the fact that significant differences are only observed in the autonomy variable. In this regard, [3] ([3]) emphasize that changes in perceived competence and the intention to engage in physical activity emerge from session 8 onward, highlighting the importance of providing sufficient time for these psychological variables to change.

It is important to highlight that the three dimensions of basic psychological needs (BPN) interact in a reciprocal manner. Satisfaction in one dimension tends to support and enhance the satisfaction of the other two ([17]; [74]; [83]). Some studies have also examined the correlation between the three dimensions of BPN ([49]; [72]). When individuals make decisions and act in accordance with their own interests (autonomy), they are more likely to engage in skill development, which strengthens their sense of competence, because the action feels self-endorsed and meaningful. Feeling accepted and connected with others (relatedness) facilitates a sense of autonomy, as it creates a safe context in which autonomy can be exercised. When someone feels competent, it becomes easier to interact positively with others. Supportive experiences of relatedness also create opportunities for learning and success, thereby further reinforcing the sense of competence ([17]; [74]; [83]).

The satisfaction of BPN can be sustained over time; however, it is neither automatic nor permanent, as it requires continuous supportive contexts and a stable internalization of values and psychological resources. In the absence of such conditions, BPN satisfaction tends to fluctuate over time ([75]). Few studies have examined BPN satisfaction over extended periods. Therefore, there is a clear need to develop longitudinal research to assess the stability and evolution of the observed effects over time. Long-term follow-ups would make it possible to identify sustained change patterns in the satisfaction of basic psychological needs and in intention to engage in physical exercise, as well as to explore the personal and contextual factors that facilitate or hinder the consolidation of these effects. In this way, the findings could substantially contribute to generating evidence on broader, more durable, and generalizable applications of intervention programs.

The satisfaction of BPN is shaped by both pedagogical approaches and cultural context. In Spain, active methodologies coexist with traditional approaches, resulting in limited autonomy and competence tied to assessment ([23]; [52]), whereas Nordic countries, Finland, the Netherlands, Canada, and Australia promote broader autonomy and competence (including problem-solving, creativity, and teamwork). In the United Kingdom (UK) and the United States (US), project-based learning combines achievement with skill development, while Asian countries emphasize self-discipline, goal attainment, and group cohesion ([11]; [13]). Relatedness also varies; Spain emphasizes close peer and teacher–student relationships ([23]; [52]), Nordic countries foster a positive emotional climate and strong social support among students, the UK and US favor peer networks with formal teacher interactions, and Asian countries emphasize group cohesion over individual ties ([11]; [13]). Therefore, the need for pedagogical approaches that are contextually sensitive and culturally adapted is highlighted.

The results support the suggestion that the satisfaction of BPN is positively related to students’ intention to be physically active, as noted by [16] ([16]). This intentionality of students to be physically active is considered a predictor of actual physical activity ([86], [85]). Satisfaction of BPN drives students to be more intrinsically motivated ([68]; [80]), developing a positive attitude that influences intention ([1]), being a predictor of behavior ([67]). This leads to the development of physical exercise habits ([77]). Therefore, increased motivation derived from improved BPN has an impact on increased participation in physical activity ([85]; [23]). The study conducted by [18] ([18]) demonstrates that SEM and GCM increase students’ satisfaction of BPN and participation, substantially contributing to the achievement of social, physical, and cognitive learning outcomes. The positive relation between BPN and intention to be physically active found in this study indicates the relevance of satisfaction of BPN in improving students’ participation in sport activities both inside and outside the school context ([85]; [23]). Appropriate regulation of students’ basic psychological needs by teachers provides educational and social benefits, as it increases the likelihood of engaging in physical activity both within and beyond the school context ([29]). This perspective is further supported by the Trans-Contextual Model of Motivation, which has been empirically validated in the Spanish context ([36]) as well as in international settings ([42]). However, the study does not analyze motivational regulations (intrinsic, extrinsic, etc.) or proximal regulators of the intention to engage in physical activity (attitude, behavioral control, etc.), so that to determine the consequences that the satisfaction of BPN generates in behaviors, we rely on the knowledge of the specialized literature. In addition, there are other aspects of great influence in the promotion of physical activity, such as the characteristics, beliefs, and involvement of families ([22]), that should be considered when assessing adherence to physical exercise.

The results indicate improvements in all the variables studied (autonomy, competence, relatedness, and intention to do physical exercise) with respect to gender. Significant differences are only shown in the relatedness variable between the genders of the students (boys > girls). The characteristics of alternative, novel, unfamiliar, playful and mixed sports ([4]; [53]) favor equal access and participation, which explains why there are no significant differences in the rest of the variables—autonomy, competence and intention between genders—as they are sports modalities that favor coeducation through appropriate equal treatment ([53]), where the role of the teacher is fundamental ([28]). These data are contrasted with the results of other research on invasion sports, which show greater satisfaction of BPN and adherence to exercise in boys ([23]). In this sense, a tendency towards unequal and stereotyped participation has been observed within invasive sports ([39]). However, alternative sports are based on more egalitarian and coeducational pillars, which can favor equitable participation. On the other hand, the study by [24] ([24]) establishes that there are no differences in the satisfaction of BPN and the intention to be physically active as a function of gender for Primary Education students. In summary, “Rosquilla” is a suitable practice to reduce gender differences in the satisfaction of BPN and the intention to be physically active. In our case, it produces differences only in the type of relatedness established between the participants. This difference can lead to decreased self-esteem and motivation in girls, which may negatively impact participation, athletic performance, and engagement, thereby exacerbating gender inequalities. Therefore, it is necessary to adjust interventions through various inclusion strategies to promote equality in girls’ sense of relatedness.

## 5. Conclusions

The results indicate that interventions employing the HM and GCM for learning an alternative sport such as “Rosquilla”, which involve the use of specific teaching techniques, styles, and strategies characteristic of each model, enhance the satisfaction of BPN. This satisfaction is essential for fostering students’ motivation, which can, in turn, lead to increased participation and a greater intention to engage in physical exercise. Therefore, the methodological approaches adopted and the teacher’s role are fundamental in promoting BPN satisfaction. Teachers should design strategies that effectively address students’ psychological needs through the appropriate use of pedagogical models, while adapting them to contextual and cultural factors.

The HM offers students greater autonomy, which is crucial for empowering self-regulation and decision-making within sports practice. This autonomy, in turn, reinforces competence, relatedness, and the intention to engage in physical activity, due to their mutual interrelation. Hence, the HM appears to be particularly suitable for school settings. Additionally, when implementing a new pedagogical model, it is important to consider the existence of a pedagogical transition period to facilitate adaptation to new instructional demands.

The heterogeneity of physical education classes is a relevant factor that must be considered when designing and implementing sports instruction. The characteristics of alternative sports, together with the proposed methodological approaches, HM and GCM, foster a more equitable learning environment that enhances the satisfaction of BPN and the intention to engage in physical exercise similarly across genders. Teachers should intentionally manage BPN within their instructional programs to promote physical, social, and health benefits by encouraging physical activity both inside and outside the school context.

### 5.1. Practical Applications

This study provides relevant insights supporting the role of the Physical Education teacher. The application of both pedagogical models, HM and GCM, may yield benefits in BPN Satisfaction and in students’ intention to engage in physical exercise. Therefore, teachers should understand and incorporate these psychological variables when designing their programs to promote student motivation and self-determined participation. Furthermore, the results demonstrate that the HM promotes a higher level of autonomy compared to the GCM, suggesting that the application of the HM may be appropriate within school contexts.

Teachers can adapt HM and GCM to different contexts by adjusting strategies to the characteristics of the group, available resources, and the needs of the athletes, considering age, experience, interests, and contextual factors such as space, equipment, and culture. To this end, it is essential to understand each model, design and manage progressive tasks, provide continuous feedback, and adapt activities to foster autonomy, competence, and relatedness in an equitable manner.

The implementation of new pedagogical models, such as HM or GCM, involves a transition period during which participants and teachers adapt to new strategies and structures; to facilitate this process and optimize its effects, it is recommended to gradually introduce the model’s elements through progressive training, provide continuous support via feedback and guidance, adapt the model to the group’s characteristics, manage tasks appropriately, and continuously evaluate autonomy, competence, and relatedness. These actions help reduce resistance to change, promote the internalization of the model’s principles, and ensure the sustainability of improvements in BPN.

The positive relationship observed between BPN and the intention to engage in physical exercise highlights the need to satisfy basic psychological needs to enhance students’ autonomous participation. It is essential to adjust inclusion interventions to ensure equality in girls’ relatedness. Key strategies include fostering girls’ integration through collaborative activities, recognizing their contributions and achievements, promoting leadership roles, assessing the school climate, and identifying gaps in the sense of belonging to implement corrective actions. Future research should explore contextual factors (age, culture, classroom climate, interests) that better explain gender differences in relatedness, as well as the implementation of measures or adjustments in interventions to ensure equality in relatedness for both genders.

### 5.2. Limitations

The reduced sample size represents one of the main limitations of this study, as it restricts the generalizability of the findings to other educational contexts or populations. A limited number of participants may increase sampling error and reduce the statistical power of the analyses, making it more difficult to detect significant effects and identify consistent patterns among variables. Likewise, the representativeness of the sample may be compromised, since participants belong to a specific educational and sociocultural environment.

Within the educational domain, each intervention and group is inevitably influenced by the characteristics of its context, creating unique and hardly replicable conditions. In this regard, it was considered necessary to respect the natural configuration of groups within the Spanish educational system, which are organized by the educational administration according to pedagogical and equity criteria to ensure an ecologically valid context. Nevertheless, this natural composition may have affected the study’s results. Therefore, it is essential to acknowledge that several factors, such as the type of sport, educational stage, individual student characteristics, the teacher’s role, institutional context, and sociocultural background, may significantly influence the results and contribute to variability in students’ responses.

Regarding contextual factors, it should be noted that each school and class group operates under its own institutional culture, available resources, teaching style, and classroom climate. These conditions directly affect the implementation and effectiveness of educational interventions, generating scenarios that are difficult to reproduce in other settings. In addition, uncontrolled variables such as students’ prior experiences or their familiarity with physical practice and pedagogical models may also have influenced the results obtained.

Another relevant limitation concerns the duration of the interventions (two months each), which may have reduced the impact of the programs. This aspect highlights the need to design longer intervention periods that allow for a progressive and stable adaptation to new educational demands. Given that the satisfaction of BPN is part of deep psychological processes, it is likely that changes in these dimensions require longer interventions to become consolidated and sustained over time. Evidence suggests that improvements in BPN satisfaction are not necessarily maintained in the long term. This consideration reinforces the need to conduct longitudinal research to examine the stability and evolution of BPN over time, as well as the personal and contextual factors that support their maintenance.

Therefore, future research should adopt extended longitudinal designs and larger, more heterogeneous samples to assess changes more accurately in psychological variables and their sustainability over time, thus enhancing the generalizability of the findings. It would also be advisable to include other relevant variables, such as motivational regulation styles (intrinsic, extrinsic) or proximal regulators of the intention to engage in physical activity (e.g., attitude, perceived behavioral control), to deepen understanding of the behavioral implications derived from BPN satisfaction. Finally, a more detailed analysis of contextual variables, such as school environment conditions, classroom climate, teacher and student characteristics, material resources, and sociocultural factors, should be conducted to better understand how these elements influence BPN satisfaction and the intention to participate in physical activity.

## Figures and Tables

**Figure 1 behavsci-15-01574-f001:**
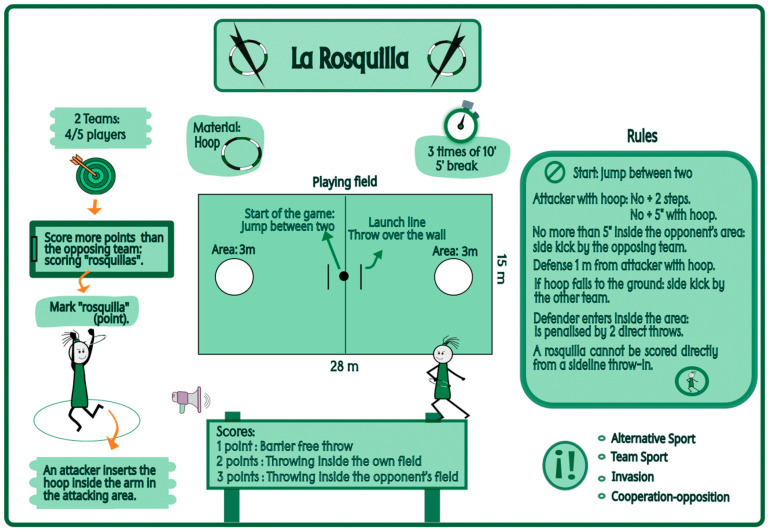
Graphic representation of “Rosquilla”. Note: Extracted and adapted from [8] ([8]).

**Figure 2 behavsci-15-01574-f002:**
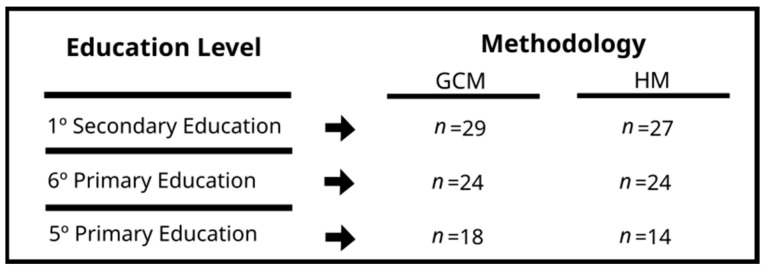
Learners’ characteristics by educational level and pedagogical model. Note: *n* = participants; GCM = game-centered model; HM = hybrid model of the game-centered model and the sports education model.

**Figure 3 behavsci-15-01574-f003:**
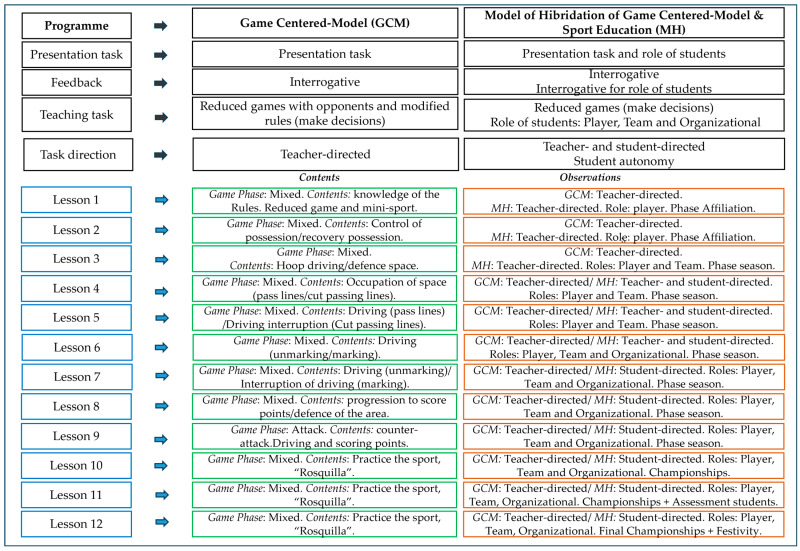
Structure of the intervention programs. Note: Extracted from [6] ([6]).

**Figure 4 behavsci-15-01574-f004:**
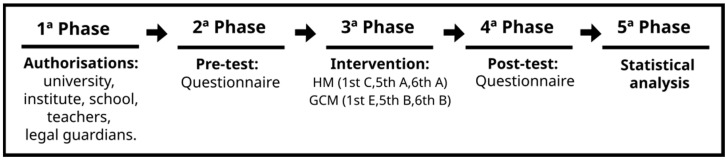
Study procedure followed. Note: Extracted from [6] ([6]). GCM = came-centered model; HM = hybrid model of the game-centered model and the sports education model.

**Figure 5 behavsci-15-01574-f005:**
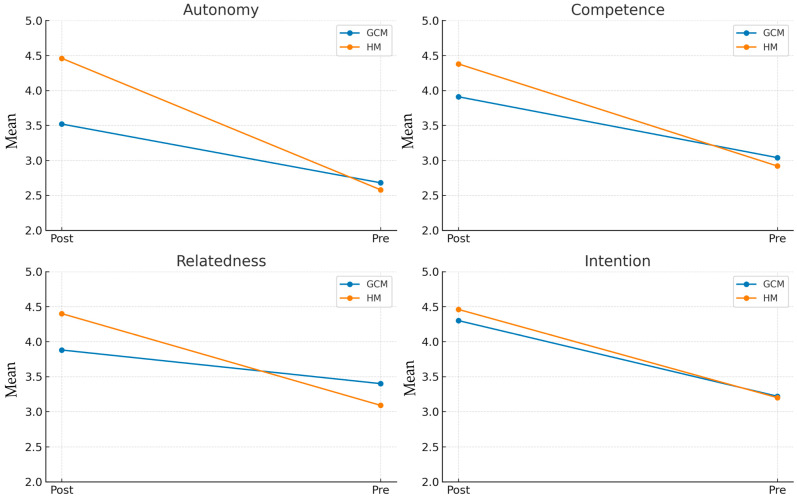
Relationship between variables (autonomy, competence, relatedness, and intention) and time points (pre-test and post-test) according to the pedagogical model.

**Figure 6 behavsci-15-01574-f006:**
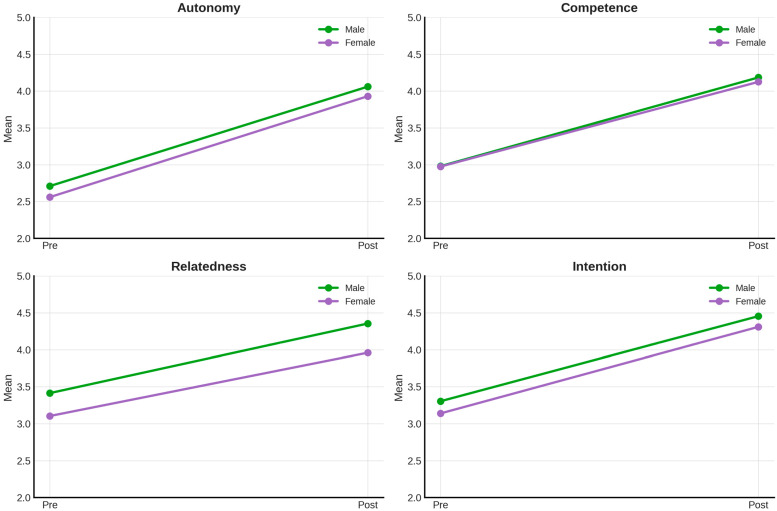
Relationship between variables (autonomy, competence, relatedness, and intention) and time points (pre-test and post-test) according to gender.

**Table 1 behavsci-15-01574-t001:** Descriptive results of the variables, BPN, and intention to exercise as a function of the pedagogical model and gender.

Program	Variables	Gender	Pre-Test *M* ± *SD*	Post-Test *M* ± *SD*	Post-Pre
GCM	Autonomy	Male	2.72 ± 0.81	3.55 ± 0.79	0.83
		Female	2.63 ± 0.72	3.49 ± 0.73	0.86
		Total	2.68 ± 0.76	3.52 ± 0.76	0.84
	Competence	Male	3.06 ± 0.77	3.86 ± 0.85	0.8
		Female	3.02 ± 0.76	3.96 ± 0.7	0.94
		Total	3.04 ± 0.76	3.91 ± 0.78	0.9
	Relatedness	Male	3.58 ± 0.67	4.05 ± 0.67	0.47
		Female	3.23 ± 0.77	3.7 ± 0.9	0.47
		Total	3.40 ± 0.74	3.88 ± 0.82	0.48
	Intention	Male	3.26 ± 0.63	4.35 ± 0.49	1.09
		Female	3.18 ± 0.70	4.24 ± 0.60	1.06
		Total	3.22 ± 0.66	4.30 ± 0.55	1.08
HM	Autonomy	Male	2.7 ± 0.56	4.57 ± 0.42	1.87
		Female	2.49 ± 0.68	4.37 ± 0.46	1.88
		Total	2.58 ± 0.63	4.46 ± 0.45	1.88
	Competence	Male	2.9 ± 0.64	4.51 ± 0.43	1.61
		Female	2.93 ± 0.69	4.29 ± 0.47	1.36
		Total	2.92 ± 0.66	4.38 ± 0.47	1.46
	Relatedness	Male	3.25 ± 0.55	4.66 ± 0.54	1.41
		Female	2.98 ± 0.82	4.22 ± 0.72	1.24
		Total	3.09 ± 0.73	4.40 ± 0.64	1.31
	Intention	Male	3.35 ± 0.71	4.56 ± 0.42	1.21
		Female	3.10 ± 0.76	4.38 ± 0.49	1.28
		Total	3.20 ± 0.75	4.46 ± 0.47	1.26

Note: Pre = pre-test; post = post-test; *M* = mean; *SD* = standard deviation.

**Table 2 behavsci-15-01574-t002:** Results of the repeated measures LMM of BPN and intention to exercise variables of the moment and model factors.

	Variables	*F*	*p*	*η* ^2^	*ϕ*	Post Hoc	Estimated Mean ± CI 95%
Intra-subject Moment (pre–post)	Autonomy	846.3	<0.001 *	0.86	1	Pre < Post	Pre: 2.63 [2.52, 2.74] Post: 3.99 [3.88, 4.10]
	Competence	477.71	<0.001 *	0.78	1	Pre < Post	Pre: 2.98 [2.86, 3.09] Post: 4.15 [4.03, 4.26]
	Relatedness	342.24	<0.001 *	0.72	1	Pre < Post	Pre: 3.25 [3.12, 3.37] Post: 4.14 [4.01, 4.26]
	Intention	1029.55	<0.001 *	0.88	1	Pre < Post	Pre: 3.21 [3.1, 3.31] Post: 4.38 [4.27, 4.48]
Inter-subject Model (HM-GCM)	Autonomy	16.2	<0.001 *	0.11	0.99	GCM < HM	GCM: 3.1 [2.96, 3.24] HM: 3.52 [3.37, 3.67]
	Competence	2.80	0.09	0.20	0.38	-	GCM: 3.48 [3.33, 3.62] HM: 3.65 [3.5, 3.8]
	Relatedness	0.84	0.36	0.001	0.15	-	GCM: 3.64 [3.48, 3.8] HM: 3.75 [3.58, 3.91]
	Intention	0.52	0.47	0.45	0.7	-	GCM: 3.76 [3.62, 3.89] HM: 3.83 [3.69, 3.97]
Interaction moment × model	Autonomy	121.6	<0.001 *	0.48	1	PostGCM < PostMH; PostGCM > PreHM; PreGCM < PostGCM; PreGCM < PostHM; PreHM < PostHM	GCM Pre: 2.68 [2.52, 2.83] HM Pre: 2.58 [2.42, 2.74] GCM Post: 2.52 [3.36, 3.68] HM Post: 4.46 [4.29, 4.62]
	Competence	30.33	<0.001 *	0.18	1	PostGCM < PostMH; PostGCM > PreHM; PreGCM < PostGCM; PreGCM < PostHM; PreHM < PostHM	GCM Pre: 3.04 [2.88, 3.2] HM Pre: 2.92 [2.75, 3.09] GCM Post: 3.91 [3.75, 4.07] HM Post: 4.38 [4.21, 4.55]
	Relatedness	74.58	<0.001 *	0.36	1	PostGCM < PostMH; PostGCM > PreHM; PreGCM < PostGCM; PreGCM < PostHM; PreHM < PostHM	GCM Pre: 3.4 [3.23, 3.57] HM Pre: 3.09 [2.91, 3.27] GCM Post: 3.88 [3.7, 4.05] HM Post: 4.4 [4.22, 4.58]
	Intention	6.27	0.013 *	0.004	0.11	PostGCM > PreHM; PreGCM < PostGCM; PreGCM < PostHM; PreHM < PostHM	GCM Pre: 3.22 [3.07, 3.36] HM Pre: 3.2 [3.05, 3.35] GCM Post: 4.3 [4.15, 4.44] HM Post: 4.46 [4.31, 4.61]

Note: GCM = game-centered model; HM = hybridization model of the game-centered model and sport education model; Pre = pre-test; Post = post-test; CI = confidence interval; *F* = statistical F of LMM; *η*^2^ = effect size; *ϕ* = observed power; * *p* < 0.05.

**Table 3 behavsci-15-01574-t003:** Control of the random effect of each LMM group for the moment and model factors.

Variables		R^2^ Marginal	R^2^ Conditional	*AIC*	*p*	*ICC*
BPN	Autonomy	0.557	0.852	571	<0.001 *	0.667
	Competence	0.438	0.767	583	<0.001 *	0.584
	Relatedness	0.305	0.798	623	<0.001 *	0.709
	Intention	0.476	0.875	527	<0.001 *	0.762

Note: BPN = basic psychological needs; *AIC* = Akaike information criterion; *ICC* = interclass correlation coefficient; * *p* < 0.05.

**Table 4 behavsci-15-01574-t004:** Results of the repeated measures LMM of BPN and intention to exercise variables for the moment and gender factors.

	Variables	*F*	*p*	*η* ^2^	*ϕ*	Post hoc	Estimated Mean ± CI 95%
Intra-Subject Moment (pre–post)	Autonomy	425.53	<0.001 *	0.76	1	Pre < Post	Pre: 2.64 [2.51, 2.76] Post: 3.97 [3.84, 4.1]
	Competence	378.63	<0.001 *	0.74	1	Pre < Post	Pre: 2.98 [2.86, 3.1] Post:4.14 [4.02, 4.26]
	Relatedness	209.91	<0.001 *	0.61	1	Pre < Post	Pre: 3.27 [3.14, 3.40] Post:4.14 [4.02, 4.27]
	Intention	970.13	<0.001 *	0.88	1	Pre < Post	Pre: 3.21 [3.11, 3.32] Post: 4.38 [4.27, 4.48]
Inter-Subject Gender (Male–Female)	Autonomy	0.84	0.36	0.01	0.15	-	M: 3.35 [3.19, 3.52] F: 3.25 [3.11, 3.4]
	Competence	0.03	0.87	0.01	0.10	-	M: 3.57 [3.41, 3.72] F: 3.55 [3.41, 3.69]
	Relatedness	8.91	0.003 *	0.01	0.10	M > F	M: 3.87 [3.71, 4.04] F: 3.54 [3.38, 3.69]
	Intention	2.15	0.14	0.002	0.08	-	M: 3.87 [3.73, 4.01] F: 3.72 [3.59, 3.86]
Interaction Moment × gender	Autonomy	0.63	0.43	0.01	0.12	Post M > Pre F; Pre F < Post F; Pre M < Post F; Pre M < Post M	M Pre: 2.71 [2.53, 2.9] F Pre: 2.56 [2.39, 2.73] M Post: 4.0 [3.81, 4.18] F Post: 3.95 [3.78, 4.12]
	Competence	0.03	0.98	0.01	0.15	Post M > Pre F; Pre F < Post F; Pre M < Post F; Pre M < Post M	M Pre: 2.99 [2.82, 3.17] F Pre: 2.97 [2.81, 3.13] M Post: 4.15 [3.97, 4.32] F Post: 4.13 [3.97, 4.29]
	Relatedness	0.02	0.93	0.1	0.84	Post M > Post F; Post M > Pre F; Pre F < Post F; Pre M < Post F; Pre M < Post M	M Pre: 3.44 [3.25, 3.62] F Pre: 3.1 [2.93, 3.27] M Post: 4.31 [4.13, 4.5] F Post: 3.97 [3.8, 4.14]
	Intention	0.27	0.61	0.16	0.31	Post M > Pre F; Pre F < Post F; Pre M < Post F; Pre M < Post M	M Pre: 3.3 [3.14, 3.45] F Pre: 3.13 [2.99, 3.27] M Post: 4.44 [4.29, 4.6] F Post: 4.32 [4.18, 4.46]

Note: GCM = game-centered model; HM = hybridization model of the game-centered model and sport education model; Pre = pre-test; Post = post-test; M = male; F = female; CI = confidence interval; *F* = statistical F of LMM; *η*^2^ = effect size; *ϕ* = observed power; * *p* < 0.05.

**Table 5 behavsci-15-01574-t005:** Control for the random effect of each LMM group of the moment and gender factors.

Variables		R^2^ Marginal	R^2^ Conditional	*AIC*	*p*	*ICC*
BPN	Autonomy	0.449	0.720	630	<0.001 *	0.492
	Competence	0.403	0.714	599	<0.001 *	0.521
	Relatedness	0.282	0.685	632	<0.001 *	0.561
	Intention	0.479	0.870	526	<0.001 *	0.750

Note: BPN = basic psychological needs; *AIC* = Akaike information criterion; *ICC* = interclass correlation coefficient; * *p* < 0.05.

**Table 6 behavsci-15-01574-t006:** Correlations between BNP and intention to exercise.

Psychological Variables		Autonomy	Competence	Relatedness
Competence	*rs*	0.644 ***	-	0.563 ***
Relatedness	*rs*	0.632 ***	0.563 ***	-
Intention	*rs*	0.561 ***	0.607 ***	0.451 ***

Note: *** *p* < 0.001.

## Data Availability

Data are contained within the article.

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
