# Peer review of "Effect of Active Pedagogical Models on Basic Psychological Needs and Intention to Do Physical Exercise"

_behavsci, 2025, doi:10.3390/bs15111574_

Round 1
Reviewer 1 Report
Comments and Suggestions for Authors
Dear authors: Congratulations on a job well done. The work is in line with the need for more applied research. However, I will point out several aspects in the introduction and the research design that need to be revised for a better result analysis.
- If you are not a physical education expert, "invasion activity" needs to be defined and characterized.
- In the introduction there is a jump between the explanation of the activity "rosquillas" and self-determination theory. It is desirable that the introduction flow more seamlessly between the variables and theories being addressed.
- As for the methodological approach, it is clear that you are using a factorial design. You can add this information, but you must be careful with the independent variables, since gender (sex) cannot be an independent variable. You can use it as a control variable or as a way to classify groups, but if you use the term “independent,” gender cannot be manipulated. So, the recommendation here is to clarify the gender as a grouping variable, but not independent.
- In table 2 and 4, is not clear the column of post hoc test. So you can find another way to present the differences between the groups; it is very hard to read.
- Lastly, the research points out the importance of physical education and improve the psychological variables associated with that. Is not clear the relation with adherence. Understanding that the population is expected to engage in physical activity when they are not participating in physical activity at school. How could these results improve adherence? I believe that the relationship established in this study is quite bold. This emphasis could be reduced.
Author Response
We greatly appreciate your comments and corrections regarding our article. We also thank you for the time you have spent reviewing this manuscript. We have carefully considered all your comments on the article.
All responses to your comments can be found in the attached file. Please consult the attached file.

Reviewer 2 Report
Comments and Suggestions for Authors
I hope this letter finds you well. I had the opportunity to review your article titled, “Effect of active pedagogical models on basic psychological needs and intention to do physical exercise". which was submitted the Behavioral sciences.
- Introduction
- Research background and problem awareness are clear: It is judged that the international context was well established by presenting the problem of decreased physical activity in children and adolescents through the World Health Organization (WHO, 2016) and related literature.
- Clarity of theoretical foundation: Self-Determination Theory and Basic Psychological Needs Theory were systematically linked, and the definitions of the three sub-factors (autonomy, competence, and relatedness) were clearly cited.
- Extensive review of prior research: It is considered appropriate to present a relatively broad theoretical background by citing numerous Spanish-language and international studies on the Sports Education Model (SEM), the Game-Centered Model (GCM), and the Hybrid Model (HM).
- However, there is a lack of a clear logical flow of the research question: Although the research hypotheses (H1-H4) are listed, there is not enough connection between how each hypothesis is logically derived from previous research.
- Lack of adaptation to an international readership: Although the “Rosquilla” event is a regional Spanish event, there are not enough visual aids or rule explanations to make it understandable to an international readership.
- Method
- Clarity of research design: Clearly describing the quasi-experimental design and pre-post tests increases confidence in the methodology.
- Compliance with ethical procedures: It is judged that transparency has been secured by specifying compliance with the Bioethics Committee approval number (159/2022) and the Helsinki Declaration (2013).
- Ensuring tool reliability and validity: By specifying the confirmatory factor analysis (CFA), Cronbach’s α, AVE, and CR values ​​for the BPNES and MIFA scales, the reliability of the survey tool and research results is enhanced.
- However, the sample lacks representativeness: the generalizability of the 136 students (elementary and middle school students) selected through non-probability convenience sampling is judged to be low.
- Incomplete randomization: Instead of group randomization, the method of “maintaining natural classes” is used, and prior differences between groups are not controlled sufficiently.
- Lack of control for teacher effects: The possibility of observer bias exists because the same researcher supervised both models. This requires further explanation.
- Results
- Highly systematic statistical verification: It is judged to be highly reliable due to the use of multi-level analysis such as CFA fit indices (CFI, TLI, RMSEA, SRMR), LMM, and ICC.
- Systematic arrangement of the results table (Tables 1–6): Pre- and post-test comparisons, gender differences, and effects between models are presented separately, improving readability.
- Presentation of effect size and power (η², Ï•): It is considered positive that the actual effect size was reported rather than simply testing significance.
- However, lack of visual data: data interpretation is difficult due to the absence of figures (e.g., change rate graphs, effect size comparison charts).
- Simplification of interpretation compared to the complexity of LMM results: Statistical indicators such as R² marginal/conditional and ICC are mentioned, but their meaning in actual educational settings (e.g., ‘proportion of individual student differences’) is not sufficiently interpreted.
- Discussion
- Clear summary of results by hypothesis: By specifying whether H1 to H4 were adopted, the logical structure is clear, which increases confidence in the research process and flow.
- Excellent comparison with previous studies: The value of this study is judged to be increased by linking the results with various studies related to SEM and GCM (Gil-Arias et al., 2017; López-Lemus et al., 2023, etc.).
- Insight into the teacher's role: The part that emphasizes 'the impact of a teacher's guidance on learner motivation and autonomy' is judged to have great practical implications.
- However, there is a lack of critical analysis: most results are interpreted only in a direction consistent with prior research, and there is a lack of discussion of conflicting results or unintended effects.
- Limitations in theoretical extensibility: Does not consider interactions between the three sub-needs of SDT or cultural differences (Spanish educational context vs. international comparison).
- Conclusion
- Consistency between conclusion and research purpose: It is judged that the research purpose (the effect of model and gender on BPN and exercise intention) and the results are consistently connected.
- Presentation of practical guidelines for teachers: The practical recommendation that HM is effective in improving autonomy is clear.
- However, the conclusion is redundant: it is judged to be at the level of repeating the summary of the Discussion and lacking emphasis on new insights or theoretical contributions.
Author Response

(The authors gave the same response as above.)

Reviewer 3 Report
Comments and Suggestions for Authors
The paper effectively examines how different teaching models in physical education impact students’ psychological needs and exercise intentions, employing a clear methodology with validated scales and robust statistical analysis. Results indicate that both models enhance autonomy, competence, relatedness, and exercise intention, with the hybrid model showing particularly strong effects on autonomy. Nevertheless, the study has some limitations: the sample is relatively small and region-specific, details on randomization and control of teacher effects are limited, the intervention period was short, and gender differences in relatedness warrant deeper discussion. The paper could also offer more practical guidance for teachers implementing these models in diverse educational settings.
Areas for Improvement:
- Sample Size and Generalizability: Include larger, more diverse samples to improve generalizability.
- Detail in Methods: Provide additional information on randomization and how teacher effects were controlled.
- Longitudinal Impact: Conduct longer-term studies to assess the sustainability of improvements.
- Gender Analysis: Explore gender differences in relatedness more thoroughly and discuss practical implications.
- Practical Recommendations: Offer clearer guidance for teachers on applying these models in various contexts.
- Abstract
- Consider specifying the age range of participants in the abstract for clarity.
- Introduction
- The introduction could benefit from a more explicit statement of the research gap and the novelty of studying the “Rosquilla” sport. For example: “There is little research on how pedagogical models affect psychological needs in alternative sports like ‘Rosquilla.’ This study is the first to explore these effects in this unique context.”
- Literature Review
- Some references are recent and relevant, but the review could be strengthened by including more international studies for broader context. Examples:
3.1. Current Situation: The literature review in the paper “Effect of active pedagogical models on basic psychological needs and intention to do physical exercise” mainly references studies conducted in Spain and a few other European contexts.
3.2. How to Improve: To broaden the perspective and enhance the relevance of the review, the authors could include international studies from other regions such as North America, Asia, or Australia. For example:
International Example 1:
Chen, A., & Ennis, C. D. (2004). Goals, interests, and learning in physical education. Journal of Educational Research, 97(6), 329-338.
This study from the United States explores how student motivation and pedagogical approaches affect learning outcomes in physical education, providing a comparative perspective on motivation theories.
International Example 2:
Dyson, B., Griffin, L., & Hastie, P. (2004). Sport education, tactical games, and cooperative learning: Theoretical and pedagogical considerations. Quest, 56(2), 226-240.
This work from Australia and the US discusses the impact of different pedagogical models on student engagement and psychological needs in PE.
- Research Questions and Hypotheses
- Consider rephrasing hypothesis 4 to clarify whether gender differences are expected for all variables or only some.
- Materials and Methods
- Provide more detail on randomization procedures and how potential confounders were controlled.
- Clarify whether the same teacher delivered both interventions and how teacher effects were managed.
- Statistical Analysis
- Justify the choice of non-parametric models more explicitly.
- Consider reporting confidence intervals for key findings.
- Results
- Highlight effect sizes and practical significance more prominently.
- Discuss the implications of the observed gender differences in relatedness.
- Discussion
- Expand on the limitations regarding sample size and contextual factors.
- Discuss the potential for longitudinal follow-up and broader application of findings.
- The authors should emphasize that future studies should include larger, randomly selected samples from multiple regions and school types. They should also consider controlling for or reporting more contextual variables (e.g., teacher training, school resources, student backgrounds).
- Conclusions and Practical Applications
- Consider specifying how teachers can implement HM and GCM in diverse educational contexts.
- Suggest ways to address the transition period when introducing new pedagogical models.
- References
- Ensure all cited works are accessible and formatted consistently.
Author Response

(The authors gave the same response as above.)

Round 2
Reviewer 3 Report
Comments and Suggestions for Authors
Dear Authors,
Thank you for your thorough and constructive responses to my comments. You addressed each point in detail and made clear improvements to the manuscript. I appreciate your positive attitude and genuine effort to enhance the quality of your work.
Best regards,
Souhail Hermassi